# MagicCartoon: 3D Pose and Shape Estimation for Bipedal Cartoon Characters

### Yu-Pei Song
Southwest Jiaotong University
Chengdu, China
yupei-song@my.swjtu.edu.cn

### Yuan-Tong Liu
Southwest Jiaotong University
Chengdu, China
yuantongliu@my.swjtu.edu.cn

### Xiao Wu*
Southwest Jiaotong University
Chengdu, China
wuxiaohk@gmail.com

### Qi He
Southwest Jiaotong University
Chengdu, China
qihe96@gmail.com

### Zhaoquan Yuan
Southwest Jiaotong University
Chengdu, China
zqyuan@swjtu.edu.cn

### Ao Luo
Southwest Jiaotong University
Chengdu, China
aoluo@swjtu.edu.cn

## Abstract

The 3D model can be estimated by regressing the pose and shape parameters from the image data of the digital model. The reconstruction of 3D cartoon characters poses a challenging task due to diverse visual representations and postural variations. This paper proposes a dual-branch structure named MagicCartoon for 3D bipedal cartoon character estimation, which models pose and shape independently through feature decoupling. Considering the correlation between category difference and shape parameters, a hybrid feature fusion technique is introduced, which integrates the global features of the original image with the corresponding local features expressed by the puzzle image, reducing the abstractness of understanding shape parameter differences. To semantically align image and geometric between feature space, a geometric-guided feedback loop is proposed in an iterative way, so that the pose of modeling results can be expressed consistently with the image. Moreover, a feature consistency loss is designed to augment the training data by incorporating the same character with different postures and the same posture of different characters. It enhances the correlation between the features extracted by the backbone network and the specific task. Experiments conducted on the 3DBiCar dataset demonstrate that MagicCartoon outperforms the state-of-the-art methods.

## CCS Concepts

• **Computing methodologies** → **3D imaging**; **Mesh models**.

## Keywords

Cartoon; 3D Pose and Shape Estimation; Feature Decoupling; Bipedal Cartoon Characters

---
*Corresponding author: Xiao Wu

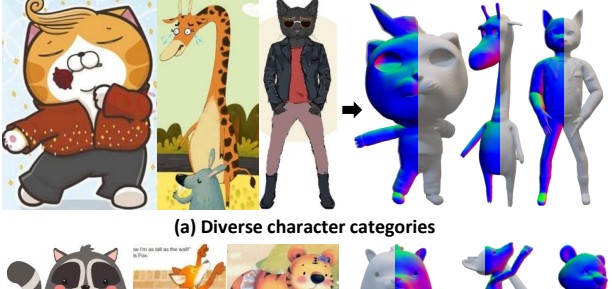

**(a) Diverse character categories**

Input image        Corresponding 3D model
**(b) Challenging posture variations**

**Figure 1: Creating a 3D cartoon model from an input image can be presented by the pose and shape parameters of the digital model. Given the vast diversity in poses and shapes exhibited by cartoon characters, this represents a challenging task that remains unexplored.**

**ACM Reference Format:**
Yu-Pei Song, Yuan-Tong Liu, Xiao Wu, Qi He, Zhaoquan Yuan, and Ao Luo. 2024. MagicCartoon: 3D Pose and Shape Estimation for Bipedal Cartoon Characters. In *Proceedings of the 32nd ACM International Conference on Multimedia (MM '24), October 28–November 1, 2024, Melbourne, VIC, Australia.* ACM, New York, NY, USA, 9 pages. https://doi.org/10.1145/3664647.3680844

## 1 Introduction

The creation of 3D models for cartoon characters is widely utilized across various domains, including video games, interactive media, and animation production. However, modeling high-quality 3D models heavily relies on skilled character designer laborious work. Deep learning can quickly create corresponding 3D models based on input images, which helps speed up this process.

Remarkable advancements have been observed in the field of digital modeling, particularly human modeling. Researchers have explored human digital models [1, 27, 34], which enable fast modeling of characters. Massive datasets of virtual and real-person scans have been exploited for deep model training. Existing methods have primarily focused on two aspects: parametric models of the human body [15, 30, 41, 46, 47] and clothing models [17, 32, 33, 38, 45]. The former estimates the shape and pose parameters of the human digital models, while the latter focuses on modeling clothed human

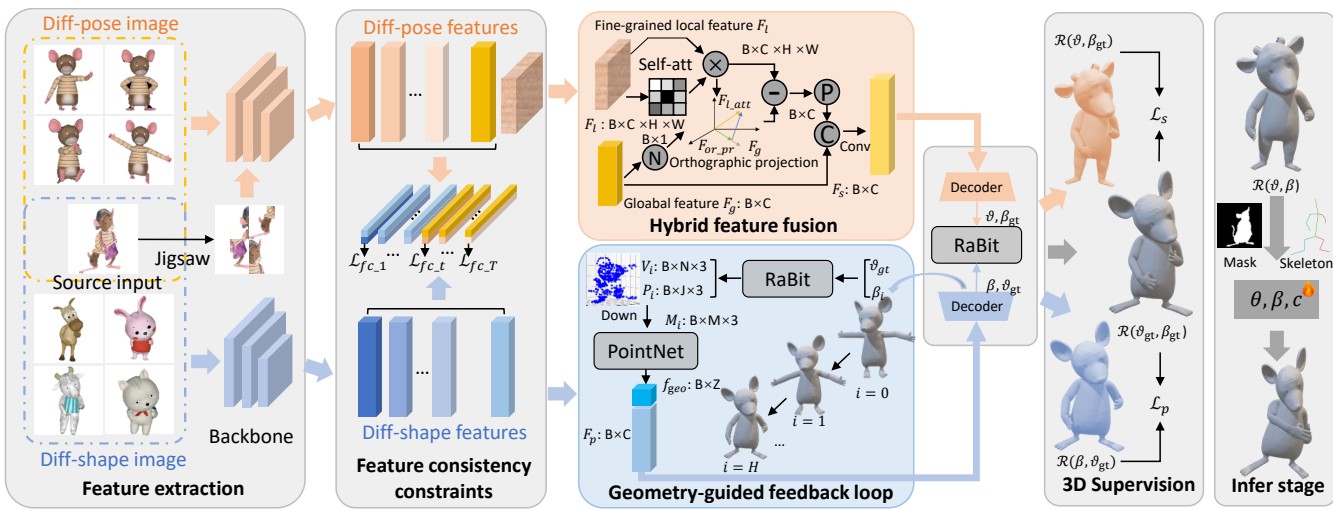

**Figure 2: The framework of the proposed MagicCartoon, which learns the pose and shape parameters of the RaBit cartoon parameter model respectively. It incorporates feature consistency constraints to different pose images sharing the same shape, as well as different shape images sharing the same pose. Additionally, the architecture leverages a hybrid feature fusion and a geometry-guided feedback loop to enhance the learning of shape and pose individually. For the in-the-wild image, the output parameters of the Rabit are fine-tuned to align with the 2D observations.**

bodies. However, 3D modeling for cartoon characters has not been fully explored. Recently, RaBit [28] introduced a novel parametric model specifically designed for bipedal cartoon characters, which have diverse appearances and postures, as illustrated in Fig. 1. The RaBit model is capable of expressing a variety of animals with distinct appearances, such as cats and giraffes. Since the reference images are mainly derived from hand-painted drawings, their poses exhibit obvious complexity.

A direct way of getting a 3D cartoon model is to train regression methods for human body parameters on cartoon data, such as HMR [18] and HybrIK [22]. However, pose and shape are two different attributes of the parametric model, and there is a conflict between their learning processes. The complex changes introduced by the diversity of cartoon characters intensify the conflict between them, thus making the learning process difficult.

To mitigate the problem of task conflict, the features corresponding to different tasks are learned independently through feature decoupling. A dual-branch structure method named MagicCartoon is introduced, illustrated in Fig. 2. This approach employs two sets of encoders and decoders to model pose and shape parameters separately. Given that cartoon characters primarily differ in elements like ears, mouth, face, hands, and feet, a jigsaw puzzle data augmentation technique is introduced. By disrupting the order of the puzzle pieces in the image, the global feature representation corresponding to the character category is destroyed to capture local details. Additionally, a hybrid fusion module is proposed to combine global features extracted from the original image with local features derived from the jigsaw puzzle-processed image. This integration enables a comprehensive understanding and representation of the input image, which reduces the abstractness of understanding shape parameters through differences in object categories. Taking into account the significant posture differences in cartoon images, a feedback loop strategy guided by geometric features is designed.

The consistency between pose modeling and image representation is achieved by iteratively reducing the differences between geometric and image information within the feature space. Moreover, a feature consistency loss is introduced to enhance the capability of the model to express relevant features. Considering improving the generalization ability of the model on in-the-wild images, 2D skeletal key points and masks of the characters are leveraged as supervisory signals during inference. This approach fine-tunes the parameters of results and achieves a better pixel-align expression with the image.

Different from traditional methods of human mesh estimation, MagicCartoon decomposes pose and shape into two distinct tasks. By eliminating task conflicts throughout the learning process, this approach facilitates extracting features that are unique to different tasks. Moreover, by incorporating the jigsaw puzzle, the network can be guided to concentrate on the detailed representation of different characters. This approach is advantageous in discriminating the shape disparities among cartoon characters. Modeling semantic errors using geometric and image features can produce accurate pose modeling outcomes.

The contributions can be summarized as follows:

- A dual-branch architecture called MagicCartoon is novelly proposed for 3D bipedal cartoon character estimation. This architecture distinctly separates the pose and shape modeling processes. It employs a feature consistency loss to enhance the capabilities of the backbone network to capture task-relevant features.
- To address the drastic shape changes of cartoon characters, a hybrid feature fusion module is proposed to produce a comprehensive expression of the input. It combines local features extracted from the input image after jigsaw puzzle processing with global features derived from the original image.
- Considering the complexity of cartoon character posture changes, a geometric-guided feedback loop strategy is introduced. This

strategy iteratively aligns the 3D modeling with the 2D input image at the feature level, minimizing discrepancies to achieve semantic alignment.

- Experiments conducted on the public 3DBiCar dataset demonstrate that MagicCartoon achieves promising performance, which outperforms the state-of-the-art approaches.

## 2 Related work

### 2.1 Human Pose and Shape Estimation

Human pose and shape estimation methods primarily fall into two categories: optimization-based methods [2, 6, 50] and regression-based methods [5, 18, 19, 23, 24, 40, 47, 49]. The early methods employed optimization to align the 2D observation data corresponding to the image by updating model parameters or vertices. The SMPL annotations of some benchmark datasets (such as the Human3.6M [16] dataset and COCO [26] dataset) are obtained through the optimization-based method [21]. However, these methods are often sensitive to initialization and may converge to local optima during iterative refinement [53]. On the other hand, regression-based methods leverage the nonlinear mapping capabilities of deep learning models. These methods directly regress the parameters of a human body model from image features. HMR [18] pioneered the approach to human mesh reconstruction, and subsequent efforts further enhanced it by leveraging the attributes of the SMPL model. PyMAF [53] minimized the interference from background information by incorporating IUV prediction. HybrIK [22] established a bridge between SMPL's 3D skeleton key points and its parameter space. However, due to the abstraction between image space and parameter space, the parameter regression method can often result in misalignment between the modeling outcomes and the original image. While the vertex regression methods [20, 25, 30, 51] offer a better solution to this issue, modeling results indicate difficulties in achieving smoothness.

Our work belongs to the parametric regression category. However, unlike existing methods, we propose a novel approach specifically tailored for cartoon characters. These characters exhibit a more extensive range of visual diversity compared to humans. A dual-branch structure network MagicCartoon integrated with feature consistency constraints is proposed. It enhances the capacity of the network for expressing features relevant to pose and shape-related tasks, while simultaneously alleviating conflicts stemming from these tasks.

### 2.2 3D Reconstruction of Cartoon Characters.

In the early stages of 3D modeling for cartoon images, 2D pictures were typically transformed into a 2.5D representation [37, 48]. When users observe the 2.5D model from varying perspectives, an interpolation algorithm is employed to generate images from the user's specific viewpoint. However, these methods do not establish a true 3D model, thereby limiting its scope of application. MagicToon [11] constructs a mesh using a single viewing angle by inputting an image mask. Photo Wake-Up [44] adopts a comparable approach, specializing in modeling human characters. It relies on the SMPL [27] model and transforms its projection to align with the mask corresponding to the input image. Nevertheless, due to the absence of 3D supervision information, these methods often lack a reasonable

structure when viewed from the side. MonsterMash [10] introduces a sketch-based 3D shape modeling technique that allows for the interactive creation of 3D shapes from sketches drawn in 2D space. Models are inflated into 3D and are easily animated from a single view. Additionally, SimpModeling [29] is dedicated to modeling the heads of animated characters. PAniC-3D [4] builds upon Nerf [31], enabling the generation of multi-view images from a single-view head image of an anime character. For the creation of complete 3D models, AvatarStudio [52] and DreamAvatar [3], combined with diffusion models [14], can achieve text-guided 3D generation of cartoon characters. However, research related to 3D conversion from 2D images to models is limited. 3DCaricShop [36] reconstructs 3D faces from 2D comics using implicit reconstruction techniques. Recently, the RaBit [28] parametric model was introduced, which constructs a cartoon character parametric model similar to SMPL. RaBit is the first 3D full-body cartoon parametric model, which creates 3D models by adjusting various pose and shape parameters. Its primary contribution lies in introducing a large-scale cartoon dataset, 3DBiCar, and a unified 3D RaBit model serving as the ground truth of 3D cartoon representation. Furthermore, RaBit proposes a BiCarNet framework that converts single-view images into 3D RaBit models. It is implemented by the regression of pose and shape parameters, which adopts the HMR [18] methodology. This advancement makes it feasible to achieve refined modeling of cartoon characters from 2D images to 3D.

Our method effectively models 3D characters that correspond to the cartoon characters depicted in the image. It integrates a hybrid feature fusion module and a geometric guidance feedback loop module to enhance the consistency of expression with the image.

## 3 MagicCartoon

### 3.1 Framework

Formally, given an input image $I$, MagicCartoon framework utilizes a dual-branch structure to model the 3D pose parameters $\beta \in \mathbb{R}^{k \times 3}$ and shape parameters $\theta \in \mathbb{R}^{d \times 1}$ of the cartoon image, where $k$ is the number of bone key points and $d$ is the dimensions of shape. The RaBit is then employed to generate the corresponding 3D mesh $\mathcal{R}(\beta, \theta) \in \mathbb{R}^{N \times 3}$ based on these parameters, where $N$ is the number of vertices. As illustrated in Fig. 2, during the training phase, the pose and shape parameters of the parameter model corresponding to the original image are changed, and the rendered image is simultaneously input into the network. This approach facilitates the decoupling of pose and shape features through feature consistency constraints. Furthermore, the fine-grained feature fusion and the geometric feature guidance feedback loop modules contribute to the regression of branch-specific parameters. In the inference stage, the framework leverages the mask and skeleton of the input image as supervisory signals to fine-tune the mesh parameters, ensuring alignment with the input image.

### 3.2 Feature Decoupling Module

MagicCartoon aims to accurately model both the shape and pose of the target character. Typically, existing methods for modeling such parameters rely on a shared backbone network, which poses a challenge as the focus areas for shape and pose modeling differ significantly. The pose parameter focuses on the posture of the

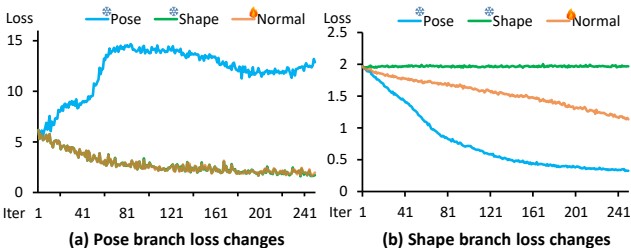

**Figure 3: The visualization diagram illustrates the evolution of vertex error loss of different parameters within a single batch, achieved by freezing the learning of a specific task during the model gradient update process.**

target character, while shape needs to distinguish the category and appearance of the target. Fig. 3 visualizes the conflicts between tasks, similar to the definition of task affinity in TAG [12]. Under the same batch, freeze the gradient generated by learning a task and then observe whether it has an impact on another task. The comparison between the blue line in Fig. (a) and the green line in Fig. (b) illustrates that once the learning of pose or shape is frozen, the parameter updates arising from the other task fail to enhance the performance of the current task. This observation indicates the presence of a conflict between the two tasks. Furthermore, training only shape parameters leads to better modeling results with lower loss values. To address this issue, a dual-branch network is designed that effectively decouples the target features from those specific to the respective tasks, enabling more precise modeling of both shape and pose.

Driven by the supervision of distinct tasks, the features learned by each branch naturally diverge. To further enlarge this divergence, a feature consistency constraint is introduced. In brief, the shape branch feature network should extract consistent features from input target images that share the same shape parameters but vary in pose parameters. Conversely, the same applies to the pose branch. This approach guides the network to prioritize task-relevant features while disregarding extraneous factors.

**Character image generator.** To generate diverse 3D models for a given cartoon character, multiple characters are randomly selected from the training set. For models exhibiting the same pose but varying shapes, the pose parameters of the current character are maintained, while the shape parameters and color maps of other characters are adopted. Conversely, to create models with identical shapes and color maps but different poses, the pose parameters are varied while keeping the shape and color consistent. RaBit is then employed to model the 3D mesh based on the pose and shape parameters. Finally, the rendering process is completed to obtain the image, as depicted in Fig. 2.

**Feature consistency loss.** The feature consistency loss is defined as the Euclidean distance measured between the feature representations of a source image and the reference image, where the latter exhibits a different pose or shape:

$$\mathcal{L}_{fc} = \sum_{i=1}^{L} \sum_{t=1}^{T} (F(I_s)_0 - F(I_{r\_i})_t)^2, \tag{1}$$

where $L$ is the number of reference images, $F$ signifies the features extracted at various stages of the backbone network, and $T$ is the total number of stages.

### 3.3 Hybrid Feature Fusion

For the human body model, besides gender, the differences between individuals are mainly reflected in height, weight, and body shape. These global features can be better captured and expressed through image extraction. However, cartoon characters exhibit differences in appearance and posture, as well as significant differences in local details, such as ears, mouths, and feet. Therefore, representing the local regions of cartoon images is equally crucial. Prior studies have demonstrated that image jigsaw puzzles are suitable for characterizing fine-grained image features [9]. The difference is that our method simultaneously leverages local features $F_l$ derived from the jigsaw puzzle and global features $F_g$ extracted from the original image. Considering the negative impact of redundant information produced by the direct fusion of two features on the quality of fused features [39]. Subsequently, the global feature $F_g$ and local feature $F_l$ are integrated into a compact descriptor through orthogonal decomposition and fusion. This process screens for key local features, ultimately providing a representative description for the shape recognition task.

**Jigsaw puzzle generator.** Given an input image $I \in \mathbb{R}^{3 \times nH \times nW}$, it is initially segmented into $n \times n$ blocks. Then, the order among these distinct blocks is randomly shuffled, and the rearranged image is forwarded to the backbone network. This process enables the network to derive the local features $F_l \in \mathbb{R}^{C \times H \times W}$.

**Orthogonal decomposition.** Before the feature fusion, $F_l$ initially establishes feature associations among diverse local blocks by leveraging the self-attention mechanism. This facilitates the recognition and selection of more crucial local characteristics:

$$F_{l_{att}} = \sigma(f(F_l) \cdot F_l), \tag{2}$$

where $\sigma$ denotes the Softplus activation function and $f$ signifies the convolution calculation involving a convolution kernel of size 1 and output channel of 1. Calculate the orthogonal components of $F_{l_{att}}$ with respect to $f_g$:

$$F_{l^*} = AvgPool(F_{l_{att}} - \frac{F_{l_{att}} \cdot F_g}{|F_g|^2} F_g), \tag{3}$$

where $| \cdot |^2$ is L2 norm calculation. The features are concatenated along the channel dimension and then processed through a layer of multilayer perceptron (MLP) to derive the final feature representation:

$$F_s = MLP(Cat(F_{l^*}, F_g)). \tag{4}$$

### 3.4 Geometry-guided Feedback Loop

The postures of cartoon characters exhibit a broader distribution than those of humans, as they often with exaggerated gestures that humans do not have, as shown in Fig 1b. Given the diversity of poses, prior works typically employ multiple iterative loops to gradually approximate the target pose from an initial one, such as HMR [18] and PyMAF [53]. A similar mechanism is designed, with the difference being the consideration of geometric feature extraction from modeling results. The advantage of this method

is that it can compare the differences between geometric features and image features at the semantic level, so as to adjust the pose parameters accordingly.

Specifically, the pose parameters and ground truth shape parameters of the i-th modeling are inputted into RaBit to derive the corresponding vertices and bone key points of the mesh. Given that the number of vertices exceeds 30k, the point cloud data are downsampled to expedite calculations and minimize redundant information [24, 53], resulting in M points. Geometric features are extracted through a 3D backbone network, which uses Point-Net [35]. The decoder establishes a connection between the image and 3D model features by integrating these features with image characteristics, outputting the pose parameter error. The entire geometry-guided feedback loop is as follows:

$$\theta_{i+1} = \theta_i + D_p(E_{3d}(Down(RaBit(\beta_{gt}, \theta_i))), F_p), \tag{5}$$

where $D_p$ represents the decoder dedicated to the pose branch, $E_{3d}$ is the 3D backbone network, $Down(\cdot)$ corresponds to the downsampling operation for the point cloud, and $F_p$ denotes the image feature of the pose branch. The symbol $i$ represents the number of iterations within the geometry-guided feedback loop. When $i$ is set to 0, $\theta_i$ is the statistical mean of the training set.

## 3.5 Training

To train MagicCartoon, a loss function is applied to the output of the model to minimize the prediction errors against ground truth, alongside the feature consistency loss for constrained learning on the backbone network. Unlike previous methods for human mesh reconstruction, only the position errors of mesh vertices and 3D skeleton points are calculated, excluding supervision of modeling parameters and 2D projection. The integration of both loss functions results in notable performance degradation. This could be attributed to the greater sensitivity of the Rabit model to parameter errors compared to SMPL, as well as the unknown camera information between the input images and the parameter model.

Let $V$ denote the output vertex position of the mesh, and $J$ is the bone joint location. The difference between the output and the ground truth is evaluated by minimizing the L1 loss:

$$\mathcal{L}_V = \frac{|V_p - \bar{V}| + |V_s - \bar{V}|}{2}, \mathcal{L}_J = \frac{|J_p - \bar{J}| + |J_s - \bar{J}|}{2}. \tag{6}$$

The error losses of the pose and shape branches are individually calculated. For the pose branch, the ground truth value of the shape serves as the input for RaBit to produce mesh vertices and bone key points. Similarly, the same approach is employed for the shape branch using the corresponding ground truth pose data as input. Our overall loss function is written as:

$$\mathcal{L} = \lambda_1 \mathcal{L}_V + \lambda_2 \mathcal{L}_J + \lambda_3 \mathcal{L}_{fc}, \tag{7}$$

where $\lambda_1$, $\lambda_2$, $\lambda_3$ represents the weight of different losses, respectively.

## 4 Experiment

## 4.1 Datasets and Evaluation Metrics

Compared to 3D human public datasets, cartoon-related datasets are relatively scarce. The 3DBiCar [28] dataset is the first extensive parametric model collection specifically tailored for bipedal cartoon characters. It has 1,500 images, which cover diverse sources ranging from book illustrations, hand-drawn artworks, computer designs, and dolls. Given an image as a reference, an artist creates a corresponding 3D model based on a template model. These models span across 15 distinct animal categories. This dataset has a unified topology similar to the SMPL model, enabling precise control through pose and shape parameters. Since the original work did not specify the distribution of training and test sets, the first 1,050 instances are set as the training dataset and the remaining ones as the test set.

The evaluation metrics are consistent with the human parameter model-based approach [22, 28], including Per-Vertex Error (PVE), Mean Per-Joint Position Error (MPJPE), and Procrustes Aligned Mean Per-Joint Position Error (PA-MPJPE). PVE measures the average Euclidean distance between the predicted and ground truth vertices, offering a comprehensive assessment of the prediction accuracy of the model. MPJPE specifically quantifies the error in joint positions, emphasizing the precision of pose reconstruction. Furthermore, PA-MPJPE focuses on local posture evaluation by excluding global rotation, translation, and scale variations. These evaluation indicators collectively provide a comprehensive analysis of the performance of the model.

## 4.2 Implementation Details

The backbone network employs ResNet-50 [13] for a fair comparison, with its parameters initialized using pre-trained weights from ImageNet [7]. The input image is cropped from the bounding box annotation. The background is eliminated based on the mask annotation to be consistent with background-free rendering image inputs. The resolution of the input image is standardized to 224×224. The quantity of supplementary input images utilized to calculate the feature consistency loss is set two. The backbone network extracts features into a 2048-dimensional vector. When processing the jigsaw puzzle image, the extracted features are 2048×7×7. The RaBit model generates mesh vertices number 38,726, along with 23 skeletal joints. After downsampling, 606 points are selected as inputs for PointNet, leading to a feature vector length of 64. The feedback loop iterates three times (H=3), and the weight distribution is set as $\lambda_1$=100, $\lambda_2$=100, and $\lambda_3$=1, respectively. The learning rate is fixed at 5e-5, the batch size is set to 32, and the jigsaw puzzle is composed of 4×4 pieces. The entire network is trained on a single RTX 3090 GPU with 250 epochs.

## 4.3 Quantitative Results

A comparison of the proposed method is conducted on the test set of 3DBiCar is listed in Table 1. Methods involving vertices are designated with ∗. The symbol † references outcomes reported in RaBit [28], while ‡ signifies the reproduction of data augmentation in this paper. To improve the generalization ability of the model, linear interpolation was carried out by the RaBit model on the 1,050 models of the training set. This is achieved by randomly selecting two models for shape and texture, and utilizing them as new model parameters. Additionally, the H3.6M [16] dataset is accessed to randomly select pose parameters, resulting in a total of 13,650 pairs of training data. The results of RaBit in Table 1 are not directly

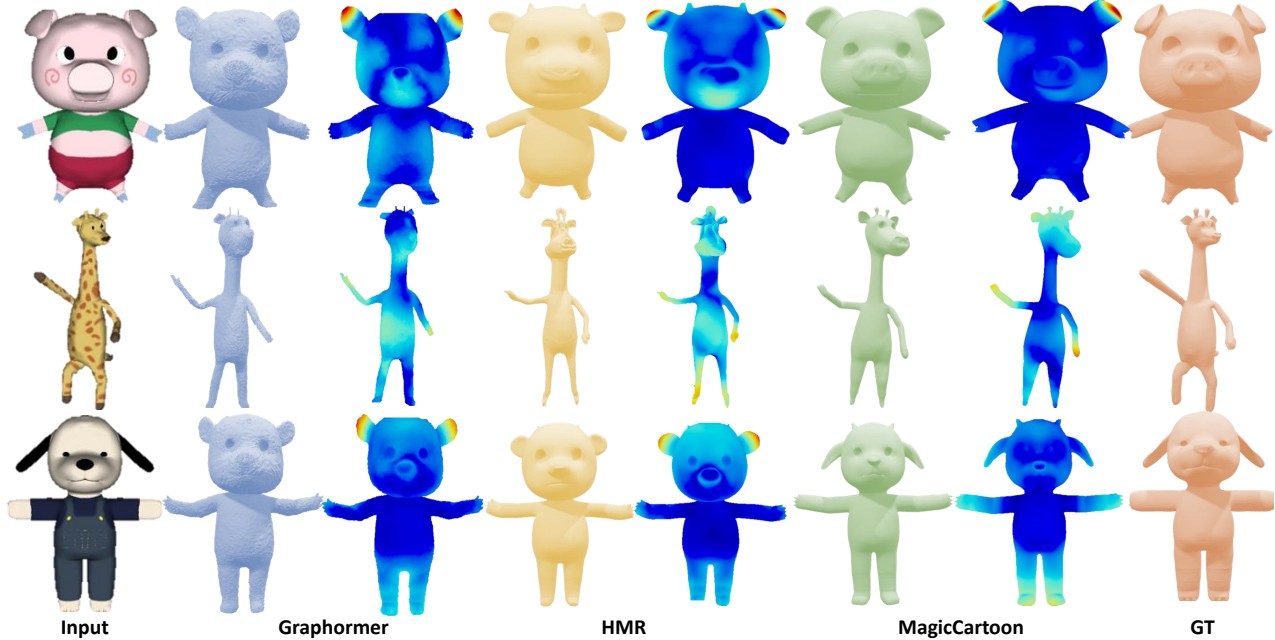

**Figure 4: Qualitative comparison with Mesh-Graphormer [25] and HMR [18] on 3DBiCar test set.**

**Table 1: Results are conducted on the 3DBiCar test datasets.**

| Type | MPVE ↓ | MPJPE ↓ | PA-MPJPE ↓ |
|---|---|---|---|
| *DecoMR[†] [51] | 85.7 | 81.2 | 47.2 |
| *Mesh-Transformer [24] | 62.5 | 49.6 | 30.7 |
| *Mesh-Graphormer [25] | 62.2 | 48.0 | 30.6 |
| HMR [18] | 65.4 | 51.3 | 30.9 |
| HMR[‡] | 60.0 | 47.8 | 28.8 |
| MagicCartoon | 61.6 | 48.1 | 30.3 |
| MagicCartoon [‡] | **56.6** | **45.2** | **28.2** |
| RaBit[†] [28] | 51.5 | 37.8 | 26.0 |

comparable. There exists inconsistency between the open-source version and its original work, which is mainly because at least 45% of the 2D images have been replaced due to copyright concerns.

Therefore, to guarantee the fairness of comparison, the aforementioned data augmentation techniques are also utilized, as listed in Table 1 with [‡]. The experimental outcomes clearly show that our method has achieved significant improvements compared to the HMR [18] baseline model used by RaBit. This is due to the design of independent modeling of pose and shape parameters for cartoon characters, which alleviates task conflicts. With the augmented data, MagicCartoon maintains superior performance. The Mesh-Graphormer [25] and Mesh-Transformer [24] for vertex regression demonstrates competitive outcomes. However, compared to SMPL, the Rabit model has a larger number of vertices, which limits its ability to achieve high-quality modeling results and leads to performance degradation.

## 4.4 Qualitative Results

The visualization results of different methods are presented in Fig. 4. The Mesh-Graphormer encounters difficulties in achieving a smooth surface due to its challenges in accurately modeling the position of a large number of vertex. Despite its superior evaluation index performance, it is not suitable for 3D modeling of cartoon characters because it requires more laborious post-processing. Furthermore, the vertex errors of the modeling results are visualized, where closer proximity to blue indicates a smaller deviation from the ground truth, and a deviation towards red signifies a larger error. In comparison to the HMR method, MagicCartoon offers more precise shape representation, since it enhances the attention to the local features of the input image. Furthermore, it facilitates obtaining accurate poses. With the assistance of geometric information, the semantic-level discrepancy between the modeling outcomes and the input image is better aligned. Despite significant pose improvements achieved compared to other approaches, there remains a wrong pose estimate. A typical failure case of MagicCartoon is shown in Fig. 5. These poses are dramatic and rarely appear in the dataset.

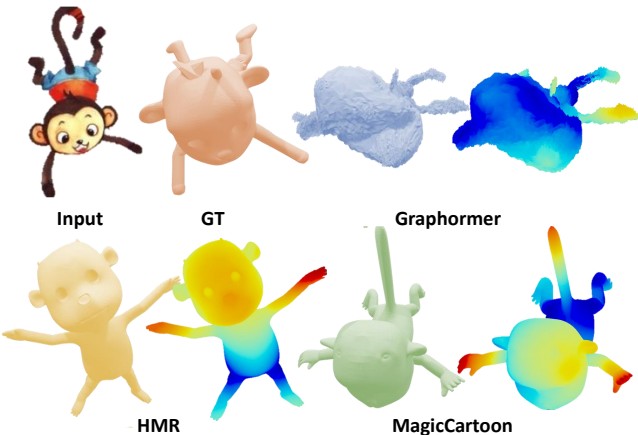

**Figure 5: Visualization of failed examples.**

## 4.5 Ablation Study

**Impact of different components**. The effect of different components is evaluated as listed in Table 2. All ablation experiments are trained on the dataset without augmentation and keep the same hyperparameter settings. The baseline method is consistent with HMR. Initially, a bi-branch structure is established, followed by the integration of feature consistency loss (**FC**). This approach enhances performance by resolving the conflicts in feature learning during pose and shape estimation tasks. Local features from the puzzle image are incorporated into the shape branch, effectively capturing local spatial information with a hybrid feature fusion module (**HF**). This refinement leads to a more robust feature representation for shape regression, further boosting the performance. A geometric feedback loop achieves optimal performance enhancement (**GFL**). Through iterative updates, the 3D model gradually attains semantic alignment with the image within the feature space.

**Table 2: Effect of individual component.**

| FC | HF | GFL | MPVE ↓ | MPJPE ↓ | PA-MPJPE ↓ |
|----|----|-----|--------|---------|-----------|
|    |    |     | 65.4   | 51.3    | 30.9      |
| √  |    |     | 63.1   | 49.6    | 31.0      |
| √  | √  |     | 63.5   | 50.0    | 30.8      |
| √  | √  | √   | **61.6** | **48.1** | **30.3**  |

**Impact of the feature decoupling module**. To assess the effectiveness of the feature decoupling module, ablation experiments are conducted on various structural designs as presented in Table 3, while maintaining the consistency of other designs.

**Table 3: Ablation experiment of the feature decoupling module.**

| Structure | FC | MPVE ↓ | MPJPE ↓ | PA-MPJPE ↓ |
|-----------|----|--------|---------|-----------|
| 1 backbone + 1 regressor |   | 65.4 | 51.3 | **30.9** |
| 1 backbone + 2 regressor |   | 65.5 | 51.4 | 31.2 |
| 2 backbone + 1 regressor | √ | 65.3 | 51.0 | 32.5 |
| 2 backbone + 2 regressor |   | 63.7 | 50.5 | 31.1 |
| 2 backbone + 2 regressor | √ | **63.1** | **49.6** | 31.0 |

The term "1 regressor" refers to the decoder head that shares the same output as the encoder. The dual-backbone network utilizes two distinct sets of image features as the input. After increasing the number of decoders, there are no significant changes in the experimental performance. This is due to the inherent independence among various neurons in MLP. Despite splitting the output, the input features remain unchanged, essentially resembling the design of a single decoder. However, when the number of backbone networks is increased, the performance of PA-MPJPE degrades significantly when using a single decoder structure. This degradation arises from the inherent conflict between the tasks, thereby restricting the learning capabilities of each task. Significant improvements in experimental performance are observed by assigning different encoders and decoders to different tasks. Furthermore, incorporating feature consistency loss further improved the performance. For intuitive comparison, a t-SNE [42] analysis is conducted of the features before and after introducing the FC loss, as well as those using a single backbone network, as shown in Fig. 6. The feature distributions across different branches of the dual-branch encoder differed significantly from the single-branch features, and the addition of the FC loss further expanded the separation between different types of features.

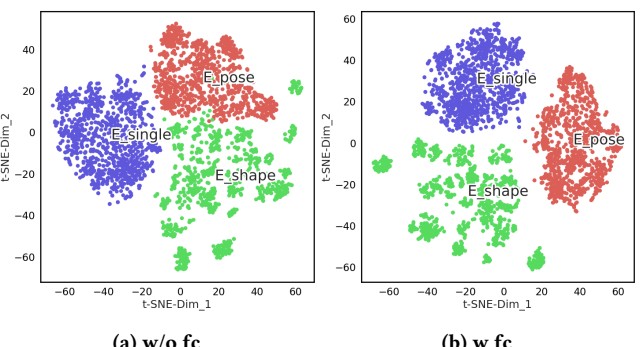

**(a) w/o fc**          **(b) w fc**

**Figure 6: The t-SNE visualization of different backbone network features.**

**The number $n$ of jigsaw patches.** An ablation experiment is conducted to investigate the impact of the number of puzzle pieces listed in Table 4. When the number of pieces is low, the improvement is minor. As the number of pieces increases, the performance is improved. However, a subsequent increase in the number of blocks results in a decline in performance. With fewer blocks, the difference between the features learned by the feature network and the original image is relatively small. As the number of blocks increases, the distance between the features extracted from the puzzle pieces and the original image increases. It becomes increasingly difficult to effectively capture meaningful features from the image. A higher number of blocks causes the entire image to look like noise, resulting in performance degradation.

**Table 4: Ablation experiment of jigsaw patch number selection.**

| Jigsaw patch number | MPVE ↓ | MPJPE ↓ | PA-MPJPE ↓ |
|---------------------|--------|---------|-----------|
| $2 \times 2$ | 63.9 | 50.5 | 31.4 |
| $4 \times 4$ | **63.5** | **50.0** | **30.8** |
| $6 \times 6$ | 63.9 | 50.5 | 31.3 |
| $8 \times 8$ | 64.4 | 50.6 | 31.4 |

**Impact of the feature fusion method.** Comparative experiments are conducted on two feature fusion methods, as listed in Table 5. The straightforward concatenation method leads to a notable decrease in performance, whereas the orthogonal decomposition approach introduced in this paper demonstrates superior performance. After orthogonal decomposition, there is no correlation between local and global features, eliminating redundant information and enhancing the local features as supplementary information. It provides a more comprehensive representation of the input image.

**Table 5: Ablation experiment of feature fusion.**

| Method | MPVE ↓ | MPJPE ↓ | PA-MPJPE ↓ |
|--------|--------|---------|-----------|
| Concat | 69.1 | 52.6 | 32.6 |
| Orthogonal | **63.5** | **50.0** | **30.8** |

**Impact of the feedback loop module.** To verify the impact of the optimal design of the feedback loop module, ablation experiments are conducted. It focuses on the integration of geometric features, selection of feature dimensions, and iteration times. The results of these experiments are presented in Table 6. The results demonstrate that incorporating geometric features, setting the feature dimension to 64, and performing 3 iterations results in optimal performance.

Given the limited number of vertices, a smaller number of feature dimensions is sufficient to represent the geometric characteristics of the model. Too many iterations may cause the model to fall into a local optimal solution. Moreover, a comparison was made of the way PyMAF [53] samples local features of the image. Specifically, features are sampled from a 14x14 feature map corresponding to 606 vertices, resulting in a feature vector of length 3,030. As camera information is unavailable, the vertices are normalized within the range of [-1, 1] to facilitate the projective transformation into the image space. However, the experimental results are unsatisfactory. For the 3DBicar data, the input image and 3D model did not pixel align. While the PyMAF approach is effective on the human-based SMPL dataset, it is unsuitable for the cartoon data associated with RaBit.

**Table 6: Ablation experiment of feedback loop module.**

| Method | Iter | Lengh | MPVE ↓ | MPJPE ↓ | PA-MPJPE ↓ |
|---|---|---|---|---|---|
| PyMAF [53] img feature | 3 | - | 67.0 | 52.7 | 32.6 |
| w/o geometry feature | 1 | - | 63.7 | 49.7 | 31.1 |
| w geometry feature | 3 | 1024 | 62.3 | 49.0 | 31.0 |
| w geometry feature | 3 | 256 | 62.2 | 48.7 | 31.4 |
| w geometry feature | 1 | 64 | 62.6 | 48.9 | 31.3 |
| w geometry feature | 3 | 64 | **61.6** | **48.1** | **30.3** |
| w geometry feature | 5 | 64 | 63.7 | 50.0 | 30.4 |

**The impact of the image background and backbone.** A segmentation module with input images at the infer stage is needed, this step is necessary due to their complicated backgrounds, which will effectively eliminate the effect of background noises, as listed in Table 7.

**Table 7: The impact of image background.**

| Preprocessing | MPVE ↓ | MPJPE ↓ | PA-MPJPE ↓ |
|---|---|---|---|
| w/o background | **61.6** | **48.1** | **30.0** |
| w background | 63.2 | 48.7 | 31.9 |

To further explore the general capabilities of the model, ablation experiments on different backbone network implementations are conducted. The ablation study results are presented in Table 8, where various backbone networks are tested. It can be seen that a larger HRNet [43] backbone network exhibits better feature representation capabilities, leading to improved performance. Despite having a similar number of parameters, the performance of the Vit [8] backbone network is not satisfactory, which merits further exploration of the underlying factors.

**Table 8: MagicCartoon with different backbone.**

| Backbone | Params (M) | MPVE ↓ | MPJPE ↓ | PA-MPJPE ↓ |
|---|---|---|---|---|
| ResNet-50 | 23.51 | 61.6 | 48.1 | 30.0 |
| HRNet-32 | 41.23 | 60.0 | 47.6 | 29.9 |
| HRNet-40 | 57.56 | 58.0 | 45.3 | 28.4 |
| Vit-S | 21.66 | 70.3 | 54.7 | 36.0 |
| Vit-B | 85.80 | 63.5 | 48.9 | 31.4 |

### 4.6 Inference for In-the-wild Images

Constructing a 3D cartoon model from in-the-wild images presents a challenge in attaining a result that harmoniously corresponds to the input image. To address this issue, the mesh vertices produced by RaBit are normalized to a value range of [-1, 1]. This normalization establishes a conversion between 3D and 2D spaces. In this process,

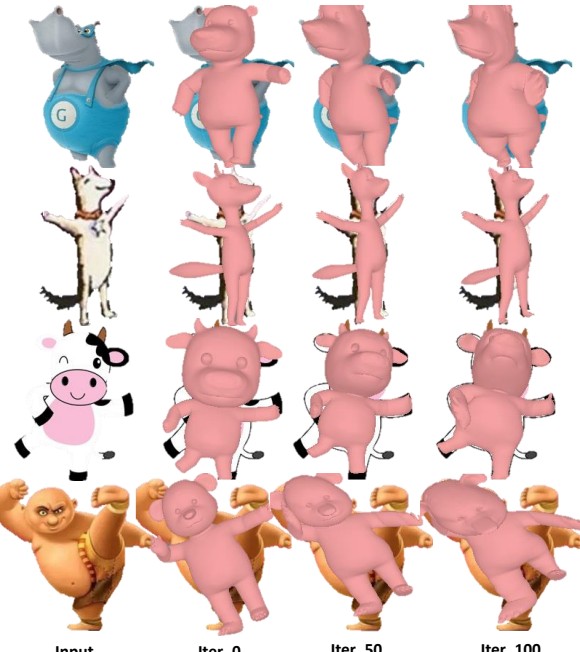

**Figure 7: Visualization of intermediate results of fine-tuning.**

$\theta$, shape $\beta$, and translation $c$ parameters as learnable variables within the Adam optimizer, aiming to minimize the loss function in the following way:

$$\mathcal{L}_{RaBit} = \min_{\theta,\beta,c}(|M_I - M_{RaBit}| + \lambda_p |K_I - K_{RaBit}|), \quad (8)$$

where $M_I$ and $M_{RaBit}$ denote the mask of the input image and render image of the RaBit model, respectively, both are optimized using L1 loss. $K_I$ represents the 2D coordinates of the bone key points. The experimental outputs are presented in Fig. 7, showing this parameter optimization technique can produce modeling results that match the input image more closely. This method depends on the initial prediction results. Therefore, it's difficult to make big changes just based on 2D observations, like in the last row of the figure.

## 5 Conclusion

In this paper, MagicCartoon is introduced as a method for cartoon character modeling. This method is designed for the complex and changeable visual characteristics of cartoon characters. The dual-branch structural design eliminates learning conflicts between pose and shape tasks. Local features from the target image are extracted through jigsaw puzzle images and orthogonally decomposed with the global features of the image to provide a more comprehensive image representation. This implicitly represents subtle differences among various characters. An iterative refinement guided by geometric cues captures semantic-level feature disparities from the input image, resulting in superior pose estimation results. Experimental results demonstrate the superiority of MagicCartoon compared to human body model methods. Additionally, the optimization of model parameters during the inference stage enhances the generalization capabilities of the algorithm for modeling diverse in-the-wild images.

# 6 Acknowledgements

This work was supported in part by the National Natural Science Foundation of China (Grant No. 62372387, 61802053), Key R&D Program of Guangxi Zhuang Autonomous Region, China (Grant No. AB22080038, AB22080039), Natural Science Foundation of Sichuan Province (Grant No. 2024NSFSC0508), Fundamental Research Funds for the Central Universities (Grant No. 2682022KJ044, 2682023ZTPY004).

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
