# OpenReview forum: "MagicCartoon: 3D Pose and Shape Estimation for Bipedal Cartoon Characters"
_acmmm.org/ACMMM/2024/Conference — MM2024 Poster_

### Official Review · Reviewer_Hxmx · 2024-05-24

**Rating:** 5
**Confidence:** 3

**Summary:**

This paper presents a method called MagicCartoon for estimating the 3D pose and shape of bipedal cartoon characters from 2D images. Due to the diverse appearance and poses of cartoon characters, directly applying human modeling methods is ineffective. To address this, MagicCartoon uses a dual-branch structure to model pose and shape independently, introduces a feature consistency loss, and employs a geometry-guided feedback loop to enhance feature representation and semantic alignment. Experimental results demonstrate that MagicCartoon outperforms existing parameter regression-based methods on the 3DBiCar dataset.

**Strengths:**

The advantage of MagicCartoon lies in its novel dual-branch structure, which independently models the pose and shape of cartoon characters, thereby avoiding inter-task conflicts. Additionally, the feature consistency loss and geometry-guided feedback loop enhance the model's feature representation and semantic alignment capabilities. Through comprehensive evaluation on the 3DBiCar dataset, MagicCartoon demonstrates superiority in reconstruction error over existing methods, proving its technical correctness and effectiveness. Its potential for generating high-quality 3D models of cartoon characters suggests broad applications.

**Limitations:**

High Technical Complexity: The implementation of MagicCartoon relies on complex feature consistency loss and geometry-guided feedback loops, increasing computational overhead and implementation difficulty, which might lead to performance bottlenecks in practical applications.

Insufficient Evaluation: The evaluation is primarily conducted on the 3DBiCar dataset, lacking tests in broader and more complex real-world scenarios, which limits its generalizability and practicality.

**Suitability:**

2

---

### Official Review · Reviewer_AHxc · 2024-05-25

**Rating:** 4
**Confidence:** 3

**Summary:**

The paper introduces a novel approach to creating 3D models of cartoon characters from single images.

**Strengths:**

1. The paper is well written and the experiments are detailed.
2. The topic is interesting and useful for VR applications.

**Limitations:**

1. The paper only shows results on the ResNet 50 backbone, I wish to see a detailed ablation study on larger backbones, e.g. HRNet 32/48, ViT series. Thus, we can measure the generalization and scalability of the proposed method.

2. How about the performance of estimation cartoon characters by using model-free methods, e.g. Mesh Transformer and Mesh Graphormer?

**Suitability:**

2

---

### Official Review · Reviewer_6paV · 2024-06-02

**Rating:** 4
**Confidence:** 2

**Summary:**

The presented work proposes a 2D cartoon character image to 3D pipeline. In the paper the authors present an architecture to estimate pose and and shape of 3D bipedal cartoon characters.

The claimed contributions are :
a novel architecture along with a different approach taking *shape* and *pose* features separately. The proposed comparisons shows good performance metrics, and the attached ablation studies support the architecture choices.

Section 2 provides related works in human pose estimation and 3d reconstruction of cartoon characters.
Section 3 present the entire framework and training procedure.
Section 4 provides several ablations studies

**Strengths:**

The presented research appears to be a complete pipeline from 2D image of character to a 3D one.

It provides comparisons against state of the art models on similar tasks on the same dataset.
INTRODUCTION and RELATED WORK sections seems to appropriately get the job done.

The central part of the paper (Section 3) goes on to described the proposed approach in detail.
The comparisons in Tab.1 show the presented model as SOTA over various metrics.

Section 4 provide a set of extensive ablation studies of the proposed model, along with a *Inference for In-the-wild Image* sub-section.
Tab(s) 2,3,4,5 are presented to justify the architectural and training choices.

**Limitations:**

LINE 8 - 30 : Minor, The abstract reads a bit weird.

The **major** weakness of the presented paper is its comparison with RaBit [27], a previous work that this research built upon but is not adequately presented in the paper.

In *Subsection 3.4 Geometry-guided Feedback Loop* the *i* index appear to be through time but is never explicitly stated.

In Tab 1. the inferior results are justified with *directly comparing the experimental results with
those provided by RaBit is not equitable due to alterations in the
dataset distribution* at lines 631-633.
Despite reading [27] it is not clear why the different performances from Tab. 1 should be justified.

The masking/segmentation Section (LINEs 858 - 909) is not explicitly clear, despite Section 4.6 (Inference for In-the-wild Image), in my opinion, there is not enough explanation on the image preprocessing part.
Does the proposed model need a segmentation module for IN-THE-WILD images?

Due the aforementioned issues the novel contributions of the paper are not clear.

**Suitability:**

3

---

### Meta-Review · Area_Chair_oq9z · 2024-06-26

**Recommendation:** Accept (Poster)
**Confidence:** 5

**Metareview:**

The paper presents a framework for recovering 3D shape and pose of cartoon characters from 2D images, which is achieved by explicitly decoupling shape and pose estimation with a dual branch network. The paper is technically sound, and the proposed solutions effective.

The paper has the following strength and weaknesses as highlighted by reviewers:

Pros:
+ A complete and effective 3D reconstruction pipeline which improves upon the state of the art
+ Extensive and convincing experimental validation
+ Potentially useful for various applications

Cons:
- High technical complexity of the framework

The three independent reviews all share a positive judgment of the paper. Most concerns raised by reviewers were related to unclear technical details, the complexity of the framework and some incomplete ablation studies. All were resolved in the rebuttal, and ultimately all reviewers agree that positive aspects overcome the negative ones, and point towards accepting the paper.

Overall, the AC agrees with the decision of accepting the paper.